# Dynamic X-ray imaging with screen-printed perovskite CMOS array

Yanliang Liu[1,7], Chaosong Gao[2,7], Dong Li[1], Xin Zhang[3], Jiongtao Zhu[3], Meng Wu[2], Wenjun Liu[1], Tongyu Shi[1], Xingchen He[1], Jiahong Wang [1], Hao Huang [1], Zonghai Sheng [4,5], Dong Liang[3,4,5], Xue-Feng Yu [1,5] ✉, Hairong Zheng [4,5,6] ✉, Xiangming Sun[2] ✉ & Yongshuai Ge [3,4,5,6] ✉

High performance X-ray detector with ultra-high spatial and temporal resolution are crucial for biomedical imaging. This study reports a dynamic direct-conversion CMOS X-ray detector assembled with screen-printed $CsPbBr_3$, whose mobility-lifetime product is $5.2 \times 10^{-4} \, cm^2 \, V^{-1}$ and X-ray sensitivity is $1.6 \times 10^4 \, \mu C \, Gy_{air}^{-1} \, cm^{-2}$. Samples larger than $5 \, cm \times 10 \, cm$ can be rapidly imaged by scanning this detector at a speed of 300 frames per second along the vertical and horizontal directions. In comparison to traditional indirect-conversion CMOS X-ray detector, this perovskite CMOS detector offers high spatial resolution ($5.0 \, lp \, mm^{-1}$) X-ray radiographic imaging capability at low radiation dose (260 nGy). Moreover, 3D tomographic images of a biological specimen are also successfully reconstructed. These results highlight the perovskite CMOS detector's potential in high-resolution, large-area, low-dose dynamic biomedical X-ray and CT imaging, as well as in non-destructive X-ray testing and security scanning.

Since its discovery in 1895, X-ray has been extensively used in many medical imaging applications such as the diagnoses and treatments of cardiovascular, pulmonary, and cancer diseases[1,2]. To provide high-quality images, usually, X-ray detectors with unique features such as high spatial resolution, rapid imaging speed, and remarkable low-dose imaging performance are highly desired[3]. Compared to the conventional indirect-conversion X-ray detectors made of scintillator materials[4], the direct-conversion detectors made of semiconductor materials offer superior X-ray imaging performance. However, the semiconductors used in the current commercial direct-conversion X-ray detectors are less satisfactory for generic medical X-ray imaging purposes. For instance, the amorphous selenium (α-Se)[5] merely works at the low energy range (<50 keV) due to its limited stopping power

(Z = 34). The cadmium zinc telluride (CdZnTe)[6] or cadmium telluride (CdTe)[7] crystals have demonstrated its advancements in resolving high-energy (>140 keV) X-ray photons, but the difficulty of growing large-scale CdZnTe/CdTe crystals makes it hard to directly manufacture large area X-ray detectors. In addition, the high cost also limits their wide spreads in medical X-ray imaging applications.

Recently, lead halide perovskites[8–13] with high X-ray absorption, high charge carrier mobility $\mu$ and long carrier lifetime $\tau$ exhibit excellent capability in ultra-high sensitive X-ray detections. For instance, Kim et al. reported the possibility of X-ray imaging with thick $CH_3NH_3PbI_3$ film on thin-film transistor (TFT) array[11] in 2017. Regardless, the integration of perovskite film with pixelated array and the reduction of dark current are still challenging. In 2021, Deumel et al.

[1]Materials Interfaces Center, Shenzhen Institute of Advanced Technology, Chinese Academy of Sciences, Shenzhen, Guangdong, China. [2]Key Laboratory of Quark and Lepton Physics, Central China Normal University, Wuhan, Hubei, China. [3]Research Center for Medical Artificial Intelligence, Shenzhen Institute of Advanced Technology, Chinese Academy of Sciences, Shenzhen, Guangdong, China. [4]Paul C Lauterbur Research Center for Biomedical Imaging, Shenzhen Institute of Advanced Technology, Chinese Academy of Sciences, Shenzhen, Guangdong, China. [5]Key Laboratory of Biomedical Imaging Science and System, Chinese Academy of Sciences, Shenzhen, Guangdong, China. [6]National Innovation Center for Advanced Medical Devices, Shenzhen, Guangdong, China. [7]These authors contributed equally: Yanliang Liu, Chaosong Gao. ✉e-mail: xf.yu@siat.ac.cn; hr.zheng@siat.ac.cn; xmsun@phy.ccnu.edu.cn; ys.ge@siat.ac.cn

developed a procedure to manufacture X-ray detector with soft-sintered $CH_3NH_3PbI_3$ on TFT array[14] by employing a grid structure to mechanically adhere the thick perovskite film. In 2022, Xia et al. prepared a TFT array via soft-pressing and in situ polymerization of multi-functional binder (TMTA) of the $CH_3NH_3PbI_3$ film[15]. Such process improves the material quality and potentially reduces the dark current.

To fully harness the benefits of perovskites in developing high-performance direct-conversion X-ray detectors, essentially, the CMOS pixel arrays[16] should be considered. CMOS arrays are dominant in consumer and prosumer imaging markets due to their distinct capabilities[17–20] to capture images with high resolution, rapid readout speed, high sensitivity, and low noise. Usually, the pixel size of CMOS can be smaller than 5 μm. In addition, specialized electrical circuits can be designed and integrated into the CMOS pixels to suppress leakage dark current. Typically, the dark current response[19] of CMOS and TFT is 200 and 2000, respectively, indicating that CMOS arrays offer superior noise immunity and lower power consumption[21]. To this end, CMOS arrays are ideal for developing cutting-edge perovskite X-ray detectors with unprecedented spatial and temporal resolution.

In this study, a direct-conversion X-ray CMOS imager made of inorganic $CsPbBr_3$ thick film (>300 μm) is developed, as shown in Fig. 1a and Supplementary Fig. 1. The prototype detector is depicted in Fig. 1b. This CMOS array (72 × 72) has a pixel dimension of 83.2 μm. An electric field of 80 V $mm^{-1}$ is applied to propel the X-ray stimulated electrons in the perovskite towards the signal collection electrode on top of each pixel. The radiographic image of a standard bar pattern (Model: Type 18-d, QUART GmbH, Germany) demonstrates that this CMOS detector can resolve fine structures above 5.0 lp $mm^{-1}$ (Fig. 1c). Moreover, a charge sensitive amplifier (CSA) is designed in each pixel to reduce the dark current via an adjustable feedback resistance[21] (Supplementary Fig. 2). As depicted in Fig. 1d, high gate voltages, denoted as $V_t$, are utilized to minimize the pixel responses (more

details can be found in Supplementary Fig. 3) for large dark current signals that primarily originates from the perovskite.

## Results

### Material and film fabrication

The inorganic $CsPbBr_3$ thick film was fabricated through screen printing process (Fig. 2a). First, the $CsPbBr_3$ precursor paste was prepared by mixing CsBr and $PbBr_2$ with equal molar mass in Dimethylformamide (DMF)/Dimethyl sulfoxide (DMSO) polar solvent through planetary ball milling (see Methods for details). The viscosity of the precursor paste varied from 110 mPa s to 9458 mPa s as the DMSO population increased (Fig. 2b and Supplementary Fig. 4). It was found that screen printing of $CsPbBr_3$ thick film can be performed when the viscosity got higher than 1500 mPa·s with DMF/DMSO ≤ 4/6. In addition, the perovskite precursor pastes gradually changed from yellow to green with bright emission (Supplementary Fig. 5), indicating a transformation from three-dimensional $CsPbBr_3$ to zero-dimensional $Cs_4PbBr_6$[22]. Afterwards, the precursor was heated to evaporate the DMF/DMSO solvent and make the dissolved CsBr and $PbBr_2$ precursors precipitated out to form the thick $CsPbBr_3$ film (Supplementary Fig. 6). Results in Fig. 2c (and Supplementary Fig. 7) demonstrate that the DMF/DMSO ratio smaller than 3/7 would benefit the growth of thick $CsPbBr_3$ films with enhanced crystallinity[23]. Regardless, excess DMSO (DMF/DMSO ≤ 2/8) may reduce $CsPbBr_3$ crystallinity due to the mixing of $Cs_4PbBr_6$ component, which has a significantly lower carrier mobility[24] and is harmful to the charge transportation and X-ray detection. Eventually, the precursor paste with DMF/DMSO = 3/7 and viscosity of 4101 mPa s were selected to screen print $CsPbBr_3$ films with adjustable sizes, shapes and thicknesses, see Fig. 2d and Supplementary Figs. 8, 9. The crystallization of $CsPbBr_3$ intermediates, which may impact the quality of the final $CsPbBr_3$ film[25], was investigated (Supplementary Fig. 10). The intermediate Cs(DMSO/ DMF)$PbBr_3$ thick film was hot-pressed under 150 °C and 0.5 MPa to rearrange the perovskite

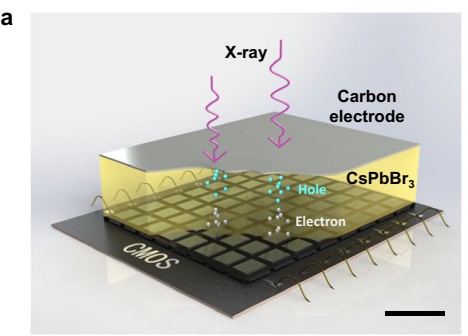

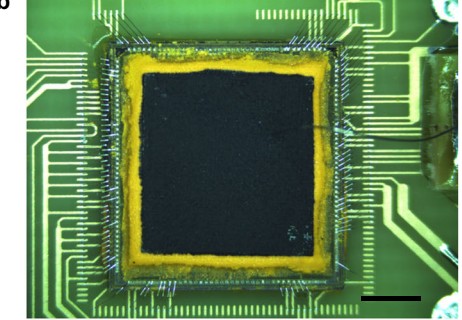

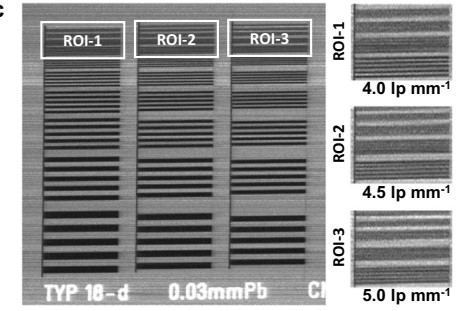

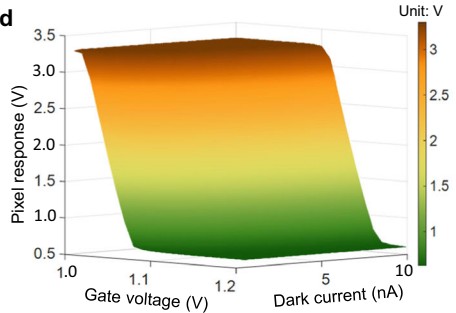

**Fig. 1 | Inorganic $CsPbBr_3$ based direct-conversion X-ray CMOS detector.**
**a** Illustration of the X-ray detector structure. The electron-hole pairs are stimulated within the perovskite film by X-ray photons. The electrons swiftly drift towards the CMOS pixels with a dimension of 83.2 μm × 83.2 μm under the applied electric field to finally generate an X-ray image. The scale bar denotes 200 μm. **b** Photograph of the fabricated X-ray detector with 6.0 mm × 6.0 mm active area. The black material

corresponds to the carbon electrode, under which is the yellow $CsPbBr_3$ film. The scale bar denotes 2.0 mm. **c** X-ray image of a standard resolution pattern. Three region-of-interests (ROIs) are selected, and the bars with 5.0 lp $mm^{-1}$ spatial resolution in ROI-3 can be distinctly visible. **d** Response map of a single CMOS pixel with respect to the gate voltage $V_t$ and the dark current. Note that the pixel response diminishes with increasing gate voltage $V_t$.

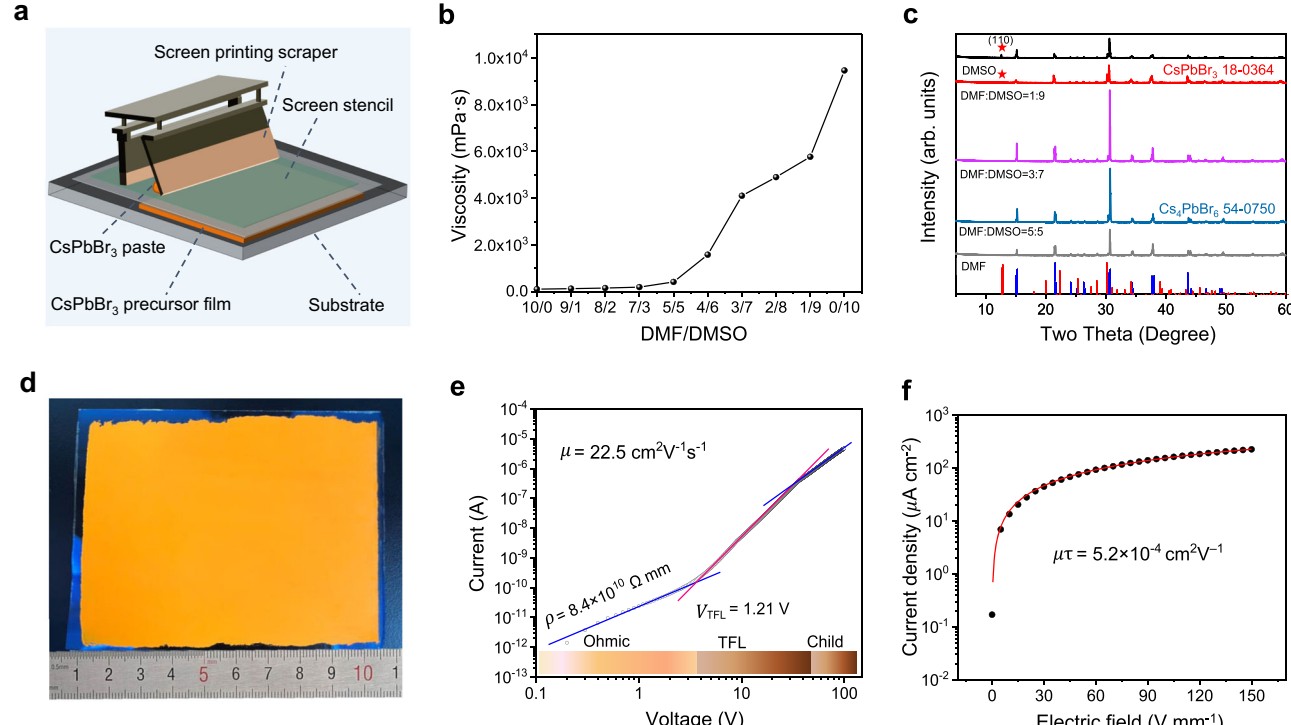

**Fig. 2 | Inorganic CsPbBr₃ perovskite thick film. a** Illustration of the screen printing setup used when fabricating the inorganic CsPbBr₃ thick film. **b** Relationship between the Dimethylformamide (DMF)/Dimethyl sulfoxide (DMSO) ratio and the viscosity of CsPbBr₃ paste. The viscosity increases from 110 mPa s to 9458 mPa s as the DMSO population increases. **c** The X-Ray diffraction analysis (XRD) spectra of the CsPbBr₃ films. The reference spectra of CsPbBr₃ and Cs₄PbBr₆ are highlighted, correspondingly. The CsPbBr₃ films exhibit enhanced crystallinity with high DMSO occupation, but excess DMSO (DMF/DMSO ≤ 2/8)

would reduce crystallinity due to the Cs₄PbBr₆ component, whose spectral location is highlighted by stars. **d** Photograph of the screen-printed CsPbBr₃ film with an approximate area of 8 cm × 10 cm. **e** The space charge-limited current (SCLC) plots. The charge carrier mobility $\mu$ is 22.5 cm² V⁻¹ s⁻¹. The trap filled limit voltage ($V_{TFL}$) is equal to 1.21 V, and the measured resistivity is equal to 8.4 × 10¹⁰ Ω mm. **f** Photoconductivity results were measured with illumination of a 460 nm LED. The mobility-lifetime ($\mu\tau$) product of the printed CsPbBr₃ film is determined to be 5.2 × 10⁻⁴ cm² V⁻¹.

grains and generate a dense and compact CsPbBr₃ thick film with improved materials crystallinity and time-resolved photoluminescence (Supplementary Figs. 11–13. According to the space charge-limited current (SCLC) measurements, the obtained CsPbBr₃ thick film exhibited a high charge carrier mobility up to 22.5 cm² V⁻¹ s⁻¹ (Fig. 2e). The mobility-lifetime product $\mu\tau$ was determined to be 5.2 × 10⁻⁴ cm² V⁻¹ (Fig. 2f). Specifically, the value of $\mu\tau$ was estimated through a curve-fitting process using the Hecht formula with respect to the collected experimental data:

$$I = \frac{I_0 \mu\tau V}{L^2}\left[1 - \exp\left(-\frac{L^2}{\mu\tau V}\right)\right], \tag{1}$$

where $I_0$ denotes the saturated current, $L$ denotes the CsPbBr₃ thickness, and $V$ denotes the bias voltage.

## X-ray response measurements
The measured current density–voltage (J–V) curves of the CsPbBr₃ detector are plotted in Fig. 3a. They showed rectifying behavior with low dark current, and the average dark-to-light current switch ratio is 219.86 with a maximum value of 1989.32 achieved at an electric field of 24 V mm⁻¹. The time-resolved current densities of the CsPbBr₃ detector were measured at varied dose rates from 686 to 4886 µGy_air s⁻¹ (Fig. 3b). As seen in Fig. 3c, this fabricated detector showed an ideal linear photocurrent response with respect to the X-ray dose rate. Quantitative liner regression analyses estimated that the CsPbBr₃ detector had a sensitivity of 3293, 8093, 15891, 25948, and 46961 µC Gy_air⁻¹ cm⁻² at electric fields of 40, 60, 80, 100, and 120 V mm⁻¹, respectively. Within a certain range of bias voltage, the

sensitivity of the detector increased proportionally (Supplementary Figs. 15, 16. The signal-to-noise ratios (SNR), defined as SNR = ($I_{photo}$−$I_{dark}$)/$I_{noise}$, were calculated (Fig. 3d). Results showed that the SNR decreased as the electric field increased. Assuming the minimum SNR = 3, the lower limit-of-detection (LoD) of the CsPbBr₃ detector was found to be 102 nGy_air s⁻¹ at 40 V mm⁻¹, 245 nGy_air s⁻¹ at 60 V mm⁻¹, and 321 nGy_air s⁻¹ at 80 V mm⁻¹, respectively (Fig. 3e and Supplementary Fig. 17). Unless otherwise stated, the comprehensive device performance of the detector was evaluated with sensitivity of 15891 µC Gy_air⁻¹ cm⁻² and LoD of 321 nGy_air s⁻¹ at electric field of 80 V mm⁻¹. To evaluate its stability, this CMOS detector was tested continuously over one-hour period. Results in Fig. 3f show a minor dark current drift of 5.2 × 10⁻⁵ nA mm⁻¹ s⁻¹ V⁻¹ and a light current drift of 3.2 × 10⁻⁴ nA mm⁻¹ s⁻¹ V⁻¹. Additional cycling tests were also performed by turning the incident X-ray beams on and off, and similar results were also obtained (Supplementary Fig. 18). The slight current drift might be caused by the field-driven ion-migration induced polarization in perovskite[26]. Low dimension and surface defect passivation strategy could be applied to restrain ion-migration, and thus enhancing the detector stability[27,28]. Additionally, the long-term irradiated CsPbBr₃ film exhibits similar X-Ray diffraction analysis (XRD) and Photoluminescence (PL) spectra compared to the original sample (Supplementary Fig. 19), indicating good irradiation stability of the fabricated CsPbBr₃ thick film.

## 2D radiographic and 3D tomographic imaging
Subsequent 2D and 3D X-ray imaging studies were performed with this perovskite dynamic CMOS detector on our benchtop system (Supplementary Figs. 20–22). Specifically, the detector was continuously

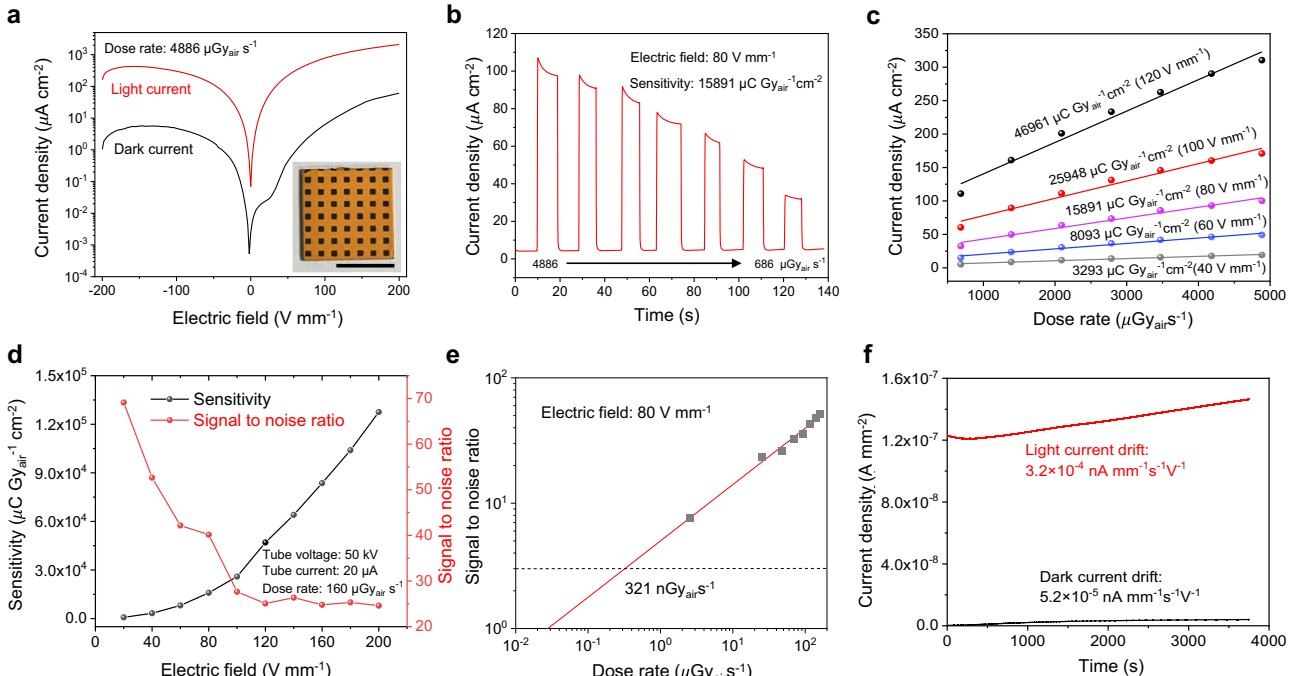

**Fig. 3 | X-ray responses of the detector. a** Current density–voltage (J–V) curve of the 15 mm × 15 mm CsPbBr₃ detector with device configuration of ITO/SnO₂/CsPbBr₃/Carbon. The dose rate is fixed at 4886 μGy$_{air}$ s$^{-1}$. The average dark-to-light current switch ratio is 219.86 with a maximum value of 1989.32 achieved at an electric field of 24 V mm$^{-1}$. The scale bar denotes 1.0 cm. **b** Photocurrent responses with respect to reduced X-ray dose rates from 4886 μGy$_{air}$ s$^{-1}$ to 686 μGy$_{air}$ s$^{-1}$. The response measurements were performed with electric field of 80 V mm$^{-1}$ and material sensitivity of 15891 μC Gy$_{air}$$^{-1}$ cm$^{-2}$. **c** The measured current densities under varied dose rates and electric fields. The corresponding material sensitivity increases as the added electric field increases from 40 V mm$^{-1}$ up to 120 V mm$^{-1}$.

**d** Variation trends of material sensitivity and signal-to-noise ratio (SNR) with respect to different electric fields. The measurements were performed under 50 kVp tube voltage and 20 μA tube current (dose rate is 160 μGy$_{air}$ s$^{-1}$). **e** The experimental lower limit-of-detection (LoD) response curve. According to the linear fitting, the LoD at the lowest signal-to-noise ratio (SNR = 3) is found equal to 321 nGy$_{air}$ s$^{-1}$ with an electric field of 120 V mm$^{-1}$. **f** Long-time stability tests of dark current and light current of the X-ray detector. Quantitatively, the drift of dark current is 5.2 × 10$^{-5}$ nA mm$^{-1}$ s$^{-1}$ V$^{-1}$, and the drift of light current is 3.2 × 10$^{-4}$ nA mm$^{-1}$ s$^{-1}$ V$^{-1}$.

scanned along the vertical and horizontal directions at a speed of 300 frames per second (fps), and 30 V external bias voltage (corresponds to 80 V mm$^{-1}$ electric field) is applied to generate the most optimal image quality. It is found that the image resolution and image SNR get balanced under this condition (Supplementary Figs. 23, 24). Moreover, the response to square-wave X-rays demonstrated a rapid rise time of 2.5 ms and fall time of 2.8 ms (Supplementary Fig. 25), allowing for quick imaging of samples larger than 5 cm×10 cm. The 2D digital radiography (DR) imaging results of an anesthetic mouse are presented in Fig. 4a, b. The two-month old mouse was scanned under two different dose levels at 50 kVp tube voltage. To compare, the same mouse was also imaged with a commercial indirect-conversion scintillator (CsI:Tl) based CMOS flat panel detector (Dexela 2329 NDT, Varex, USA). As seen from the zoomed-in region-of-interests (ROIs) in Fig. 4b, the skull, ribs and pelvis of the mouse can be clearly delineated on the DR images obtained from this direction-conversion perovskite CMOS detector, demonstrating better spatial resolution to the indirect CMOS detector. Moreover, our detector outperforms the indirect CMOS detector even with 50% less radiation dose, indicating its better capability for low-dose X-ray imaging. Additionally, a chicken drumette specimen was tomographically scanned to validate the 3D CT imaging performance of such perovskite CMOS detector (Fig. 4c). In total, 180 projections were acquired in a sequence by rotating the specimen continuously with two-degree angular interval (see Methods for details). CT images were reconstructed using the standard Feldkamp-Davis-Kress (FDK)[29,30] algorithm with a Ramp filter. As seen in Fig. 4c, the bony structures and the tissues can all be clearly identified. A 1.0 mm thick tungsten plate (Supplementary Fig. 26) was imaged to calculate the modulation transfer function (MTF)[31]. In

addition, the normalized noise power spectra (NPS)[32] of the detector was also characterized (Supplementary Fig. 27). The measured MTF and NPS curves are plotted in Fig. 4d. The nearly flat NPS curve suggests minimal signal correlation (cross-talking) between neighboring pixels. More imaging results of different objects are available in Supplementary Figs. 28, 29.

In this study, a high-performance perovskite based direct-conversion X-ray CMOS imager was developed. Results demonstrated that the screen-printed thick CsPbBr₃ film has a high $\mu\tau$ product of 5.2 × 10$^{-4}$ cm$^2$ V$^{-1}$, a high X-ray detection sensitivity of 15891 μC Gy$_{air}$$^{-1}$ cm$^{-2}$, and a low dose detection limit of 321 nGy$_{air}$ s$^{-1}$. Experimental X-ray 2D imaging results demonstrate that this perovskite CMOS detector can achieve very high spatial resolution (5.0 lp mm$^{-1}$, hardware limit is 6.0 lp mm$^{-1}$) and low-dose (260 nGy) imaging performance. Moreover, 3D CT imaging was also validated with this detector at a fast signal readout speed[33] of 300 fps.

The use of lead halide perovskites offers a promising blueprint for developing the next generation state-of-the-art X-ray detector with significantly enhanced spatial resolution, readout speed and low-dose detection efficiency. As a consequence, conventional medical X-ray imaging would become more gentle and safe in the near future. This technique is particularly promising for medical X-ray imaging applications such as dental imaging and breast imaging with a linear perovskite CMOS array or a large area perovskite CMOS flat panel.

## Methods
### Ethics Statement
The imaging experiments with a mouse in this study adhered to ethical guidelines set by the Regional Ethics Committee for Animal

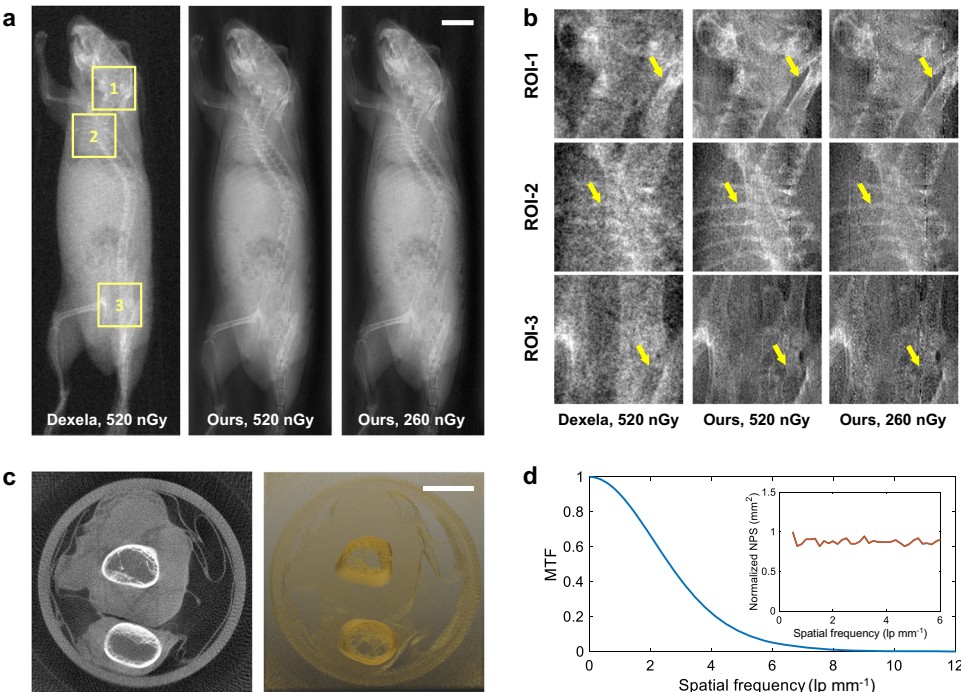

**Fig. 4 | X-ray imaging results. a** The radiographic imaging results of a mouse. Image in the first column is obtained from a commercial indirect-conversion CMOS X-ray detector with 74.8 μm pixel size. Images in the second and third columns are obtained from our direct-conversion perovskite CMOS X-ray detector with 83.2 μm pixel size at two different dose levels: 100% dose and 50% dose. The scale bar denotes 2.0 mm. **b** The zoomed-in region-of-interests (ROIs). As seen, the skull, ribs, and pelvis images obtained from the developed direct-conversion CMOS detector are much sharper and less noisy. Moreover, our detector outperforms the indirect conversion CMOS detector even with 50% less radiation dose. **c** The reconstructed CT images of a chicken drumette specimen. Image on the left represents a single image slice, and the image on the right renders the entire 3D CT volume. The scale bar denotes 2.0 mm. **d** The measured modulation transfer function (MTF) response of the perovskite CMOS detector. The 10% MTF corresponds to about 5.0 lp mm$^{-1}$. The normalized noise power spectrum (NPS) response of the perovskite CMOS detector is plotted in the upper left corner. The flat NPS curve indicates a negligible signal correlation between neighboring pixels.

Experiments and the Care Regulations approved by the Institutional Animal Care and Use Committee (SIAT-ACUC-231108-YGS-SZH-A2338) of Shenzhen Institute of Advanced Technology, Chinese Academy of Sciences. A black mouse (Strain: C57BL/6JNifdc; Stock #: 219; Age: 6-8 weeks; Gender: male) was procured from Beijing VENTOLIN Experimental Animal Technology Co., Ltd. Pentobarbital sodium salt (Sigma-Aldrich, 1% concentration, dosage of 150 μl per 20 g body weight) was administered via intraperitoneal injection to induce profound anesthesia (duration of 30 minutes) in the mouse 10 minutes prior to the X-ray imaging with the perovskite CMOS detector.

## Materials
PbBr$_2$ (lead bromide, 99%, Aladdin), CsBr (cesium bromide, 99.5%, Aladdin). Organic solvents including dimethyl formamide (DMF, AR, 99%) and dimethyl sulfoxide (DMSO, AR, 99%) were purchased from Sigma-Aldrich. Equivalent molar ratios of CsBr and PbBr$_2$ made up to be 80 wt% were mixed into DMF/DMSO solvent, and then the mixture was grinded for 6 hours to obtain the viscous CsPbBr$_3$ precursor paste for screen printing.

## Device fabrications
The substrates (glass/ITO or CMOS) were cleaned with detergent, ultra-sonicated in acetone and isopropyl alcohol, and subsequently dried overnight in an oven at 100 °C. For the fabrication of SnO$_2$ film, the colloidal SnO$_2$ aqueous dispersion (15 wt%) was diluted using deionized water to 5 wt%, and the obtained SnO$_2$ solvent was spin-coated on the substrate (4000 rmp, 40 s), which was annealing at 140 °C for 1 hour, to form an electron transport layer[34]. Next, the viscous CsPbBr$_3$ precursor paste was screen-printed on the substrate for several times to obtain a CsPbBr$_3$ precursor film with desired thickness. The CsPbBr$_3$ precursor film was pre-heated at 100 °C for 10 min, resulting an intermediate CsPbBr$_3$ thick film with trace amount of residual DMSO. Afterwards, such intermediate CsPbBr$_3$ film was hot-pressed at 150 °C and 0.5 MPa for 10 min. The hot-pressed CsPbBr$_3$ thick film was further heated at 150 °C for 30 min to obtain a complete CsPbBr$_3$ thick film. At last, a carbon paste was screen-printed on top of CsPbBr$_3$ thick film and then heated at 120 °C for 30 min to form a top carbon electrode.

## CMOS array and signal readout
The CMOS array consists of 72 × 72 active pixels, a scan module, a switch array and a buffer (Supplementary Fig. 2). The analog signal of each pixel is transmitted through the switch array and the buffer. The switch array is controlled by the scan module. The rolling shutter readout scheme is adopted in the CMOS array and the maximum signal readout speed can be higher than 300 frames per second. The charge collection electrode, denoted as T, is an exposed top-most metal. Another top-most metal, denoted as G, covered by an insulation material is used to form a focused electric field to improve the charge collection efficiency. The readout circuit of each pixel contains a charge collection electrode, a charge sensitive amplifier (CSA) and a two-staged cascaded source follower. The charge collected by the collection electrode is fed into the CSA and then converted into a voltage signal. The gate voltage of the feedback transistor is adjusted according to the decay time of the CSA to maximize the output dynamic range.

## Data acquisition
The signal readout circuit is composed of a chip binding board, a mixed signal converter board and a Field-Programmable Gate Array

(FPGA, Xilinx kintex-7, AMD, USA) control board (Supplementary Fig. 22). The binding board carries a bare die of the CMOS array, and is connected with the mixed signal converter board through a pin header. Filtering capacitors are placed on the binding board to suppress noise. The single-end analog signal output from the CMOS array is fed into the mixed signal converter board and then converted to a differential signal to match the input of the analog-to-digital converter (ADS5282, TI, USA), which features a 12-bit digital resolution, a sampling rate of 65 MSps and an interface of the serialized Low Voltage Differential Signaling (LVDS) outputs. A 16-bit digital-to-analog converter (DAC8568, TI, USA) on the mixed signal converter board is designed to configure the bias voltages of the CMOS array. The mixed signal converter board is connected with the FPGA based control board through FPGA Mezzanine Card (FMC) connector. The FPGA is the core logic control in a scalable and minimum hardware including an Ethernet transceiver module, two Double-Data-Rate Three Synchronous Dynamic Random Access Memory (DDR3 SDRAM) module and a flash configuration module. The FPGA based control board receives the serialized data output from the analog-to-digital converter and transmits them to PC via Ethernet.

### X-ray imaging and characterizations

The in-house-built X-ray CT imaging benchtop had a medical-grade X-ray tube (G-242, Varex, USA). The tube current was fixed at 12.5 mA during the continuous exposures. A beam collimator was utilized to regulate the beam shape. The fixed beam filtration was 1.5 mm aluminum. The distance between the X-ray source and the scanned object was 444.0 mm, and the distance between the X-ray source and the CMOS detector was 476.0 mm. The horizontal linear stage (LS12-X200, Hanjiang, China) had a travel distance of 200.0 mm, and the vertical linear stage (STS06-X20, Hanjiang, China) had a travel distance of 40.0 mm. For CT imaging, the object was scanned on a rotation stage (Model: URS100BCC, Newport, USA) by 360 degrees with two-degree interval. The MTF and NPS experiments were measured with RQA3 beam (50 kVp tube voltage, 10.0 mm aluminum filtration). The CMOS detector array was aligned with the X-ray source. The 1.0 mm thick tungsten (W) plate was tilted by 3.0 degrees on the detector surface.

### Image correction and stitching

Gain correction (Supplementary Fig. 24) was performed before stitching the thousands of X-ray projections into a full DR projection image. In particular, the signals corresponding to the same location were added together and the signals from different locations were rearranged sequentially to obtain the final images.

### Reporting summary

Further information on research design is available in the Nature Portfolio Reporting Summary linked to this article.

## Data availability

All relevant data generated in this study are provided in the paper and its Supplementary Information/Source Data file. The source data has been deposited in Figshare database under the accession link of https://doi.org/10.6084/m9.figshare.24922572.

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

## Acknowledgements

This work was financially supported by the Shenzhen Basic Research Program (JCYJ20200109115212546, Y.G.), National Natural Science Foundation of China (12235006, X.S., 12027812, Y.G., 62004091, Y.L.), Shenzhen Science and Technology Program (JSGGKQTD20210831174329010, H.Z., ZDSYS20220527171406014, D.Liang), Youth Innovation Promotion Association of Chinese Academy of Sciences (2021362, Y.G.).

## Author contributions

Y.L., D.Li, W.L., and T.S. prepared the CsPbBr3 film and made the characterization measurements. C.G. and X.S. designed the CMOS array. X.Z., J.Z., M.W., and Y.G. performed the data acquisition and image analyses. Y.L., C.G., and Y.G. analyzed the results and wrote the manuscript. X.H., J.W., and H.H. prepared the measuring setup. Z.S., D.Liang, H.Z., and X.-F.Y. joined the final discussion. Y.G., X.S., X.-F.Y., and H.Z. supervised all the activities.

## Competing interests

The authors declare no competing interests.
