## [Peer Review File · Nature Communications]

Dynamic X-ray imaging with screen-printed perovskite CMOS arrayREVIEWER COMMENTS

Reviewer #1 (Remarks to the Author):

This article presents a dynamic X-ray imaging array by integrating the screen-printed perovskite thin film with a CMOS array. The high-performance X-ray detector array was achieved by optimizing the fabrication process of perovskite, exhibiting rapid dynamic imaging response (300 frames). This work focuses on technical optimization, but the specific performance parameters do not outperform those of documented works. The following questions need to be addressed first before accepting the manuscript for publication in Nature Communications.

1. In the abstract, the authors stated “low-dose (50% less) imaging capability” without specifying the specific comparison object and X-ray dose. This could cause misunderstanding.
2. Why does a higher gate voltage lead to a lower dark current (Fig. 1d and Supplementary Fig. S3)? More relevant explanations, circuit diagrams, or device working principles need to be provided.
3. The lower limit-of-detection (LoD) in this work was 346 nGyair s⁻¹ at 80 V/mm. No obvious advantage could be observed when compared with published works on similar perovskite materials.
4. In addition, the value of LoD in this work is estimated by the signal-to-noise ratio (Fig. 3e), but the X-ray dose used in the test differs from the estimated value by more than two orders of magnitude. Please provide more experimental data tested at low X-ray doses as support to increase credibility.
5. This detector can obtain a fast signal readout speed of 300 fps in this work. The response time of a single-pixel detector should be provided. Moreover, relevant data or reference on the performance of circuits and devices that enable rapid acquisition is also required to be provided.
6. The array size and pixel density are very low (72×72 pixels, 6×6 mm²). This overshadows advantages of the CMOS array.
7. The author mentions that “the stimulated electrons inside the perovskite film were nicely confined within a single pixel by the electric field.”. In my opinion, the carriers generated in adjacent perovskite pixels are more likely to be attracted under larger electric fields (80 V/mm) because perovskite materials have long carrier lifetimes and carrier diffusion lengths.
8. Why was carbon paste used as the electrode material?
9. In Fig. 3b, why does the photocurrent rise first and then fall?
10. In Fig. 3e, there is a typo, “Signal to noise ratio”.

Reviewer #2 (Remarks to the Author):

The authors present an interesting X-ray detector, combining the all inorganic CsPbBr₃ perovskite and complementary metal-oxide-semiconductor (CMOS) pixels. Although the device does not exhibit record sensitivities or mobility lifetime products, they are among the best in perovskite-based devices. Moreover, they show exceptional 2D and 3D imaging utilizing the CMOS array, lowering the needed dose-rate and increasing spatial resolution, all at a fast readout speed.

They definitely emphasize the potential of perovskite when combined with CMOS for medical

applications.

The study is well designed and presented. Moreover, the manuscript is nicely written and reads well.

I have a few questions, that if answered, I would recommend publication in Nature Communications.

- 1) When discussing the screen-printing of CsPbBr₃, could you elaborate a bit more on the intermediate phase, which is common in perovskite in similar printing methods. (<https://doi.org/10.1021/acsnano.0c07993>)
- 2) Indicate the light source used in your mobility lifetime calculations.
- 3) You mention the maximum dark to photocurrent ratio is 2000, however this is at a position of a dip in the dark current. Why is the dip? Is it reproducible? An average ratio would be better to use. (Fig. 3a)
- 4) You varied your dose-rates from higher to lower, did you do any reproducibility checks and have you started with lower to higher dose rates. Can you explain the initial drop in the photocurrent, which eventually starts to rise as seen in Figure 3f. What can be done to keep this current stable for application, even in the cycling test the baseline is moving? I believe this should be discussed a bit more in detail.
- 5) Did you characterize your sample after these long-term tests (cycling or under constant operation/irradiation), any changes or degradation?

Reviewer #3 (Remarks to the Author):

This manuscript is written about X-ray detectors with CMOS array and has been systematically studied. Therefore, the reviewer believes that the manuscript is suitable for publication in Nature Com., but he believes that further consideration is necessary for the following points.

1. In the background, the authors state that "the pixel size of CMOS arrays can be easily made less than 5 μm ". However, the pixel size of the actually utilized CMOS array is about 80 μm . At the moment, this seems to be the limit when considering the crystal size, etc., but if there is any other reason for using this size CMOS array, please mention it.
2. The advantage of CMOS arrays, suppression of leakage dark current, is also mentioned in the background. This is interesting for readers. Please describe about the dark current results for the CMOS array (with special electrical circuits) and the TFT array.
3. Regarding the J-V curve in Fig.3.a, (i) Dark current is usually determined by the resistivity of the film and the contact with the electrode. Light current is determined by detection sensitivity and quantum efficiency in addition to dark current. So each should have a different electric field dependence, but why do they look nearly translated?
(ii). Related to the previous question, why is the electric field not monotonically decreasing over the -190--170 V/mm range? The reviewer is concerned that the electric field may not simply represent the electric field in the detection layer. (For example, the electric field is applied to the electrode or the interface between the electrode and the detection layer more than expected.

4. Please add measurement conditions (tube voltage, tube current, dose rate) about Fig 3.d.
5. For Fig3.e, it is unacceptable to determine the detection limit with this extrapolated line. If this film can detect 1/1000 of the intensity of X-rays that had actually been detected, show that X-rays of that dose can be detected by adjusting the dose rate with an attenuation plate or the like.
6. As for the experiment on Fig. 4, the reviewer thinks the results are excellent. However, if there is a reason why the result shown is only at 80V/mm, please describe. For example, a measurement of 20 or 40 V/mm, which has a higher SNR, should give a higher-resolution image.
7. Please briefly explain the necessity of SnO₂ for the methods (device fabrication). The authors can also reference some literature.

Title: Dynamic X-ray imaging with screen-printed perovskite CMOS array

Paper ID: NCOMMS-23-31690-T

We would like to sincerely thank the reviewers and editors for their time and efforts in reviewing our manuscript. All comments and suggestions have been addressed carefully. Please find our point-by-point responses below.

Note: the reviewers' comments are in blue italic text, and our point-by-point responses are in plain black text.

Reviewer #1 (Remarks to the Author):

This article presents a dynamic X-ray imaging array by integrating the screen-printed perovskite thin film with a CMOS array. The high-performance X-ray detector array was achieved by optimizing the fabrication process of perovskite, exhibiting rapid dynamic imaging response (300 frames). This work focuses on technical optimization, but the specific performance parameters do not outperform those of documented works. The following questions need to be addressed first before accepting the manuscript for publication in Nature Communications.

Authors' response: Thank you.

1. In the abstract, the authors stated "low-dose (50% less) imaging capability" without specifying the specific comparison object and X-ray dose. This could cause misunderstanding.

Authors' response: Thank you. We are sorry for such misleading description. The low-dose imaging capability of this perovskite CMOS detector is demonstrated by comparing with the conventional indirect CMOS X-ray detector, as depicted in Fig. 4. To clarify, it has been updated as follows: Compared with the conventional indirect CMOS X-ray detector,

DR imaging results show that such perovskite CMOS detector is able to perform high spatial resolution (5.0 lp/mm) X-ray imaging at low radiation dose level (260 nGy). Please see the revised manuscript on Page 1.

2. Why does a higher gate voltage lead to a lower dark current (Fig. 1d and Supplementary Fig. S3)? More relevant explanations, circuit diagrams, or device working principles need to be provided.

Authors' response: Thank you. For our designed CMOS, the dark current (I_d) is assumed to be fixed under a certain external electric field. Moreover, the output baseline voltage V_o , which is expressed as

$$V_o = I_d \times \frac{1}{\mu_n C_{ox} \frac{W}{L} (V_G - V_S - V_{th})},$$

where I_d denotes the dark current, μ_n denotes the electron mobility, C_{ox} denotes the gate oxide capacitance per unit area, W and L denote the width and length of the feedback transistor, respectively, V_G denotes the gate voltage of the feedback transistor, V_S and V_{th} denote the source voltage and the threshold voltage of the feedback transistor, respectively, both of which are fixed in the front-end circuit. Therefore, the output baseline voltage V_o decreases as the gate voltage V_G increases. Please see the revised Supplementary Fig. S3.

In particular, a Charge Sensitive Amplifier (CSA) with a suppressing dark current circuit used in the CMOS array is shown below. The dark current from the detector is absorbed by the transistor M1. The current generated by M1 is automatically adjusted according to the feedback circuit which is composed of a resistor, a capacitor and an open-loop amplifier to match the dark current of the detector. Hence, the baseline of the CSA_OUT node can be obtained with the same level of the CSA_VREF node. As a consequence, the output dynamic range of the CSA is not affected by the dark current.

A charge sensitive amplifier (CSA) with a suppressing dark current circuit used in the CMOS array.

3. The lower limit-of-detection (LoD) in this work was $346 \text{ nGy}_{\text{air}} \text{ s}^{-1}$ at 80 V/mm . No obvious advantage could be observed when compared with published works on similar perovskite materials.

Authors' response: Thank you. Admittedly, the detection limit is an important figure of merit for evaluating the imaging performance of any X-ray detector. Always, the lower limit-of-detection (LoD) is desired. This is also held for X-ray detectors made of perovskite materials.

To compare, key parameters of direct perovskite X-ray detectors reported in previous literature are summarized in the following table. As seen, the low limit-of-detection (LoD) ranges from 0 up to $500 \text{ nGy}_{\text{air}} \text{ s}^{-1}$, depending on the type of perovskite material. For example, detectors made of single crystal perovskite often have extremely low LoD down to a few $\text{nGy}_{\text{air}} \text{ s}^{-1}$ because of the high crystallinity and low defect density of the special material. However, detectors made of thick polycrystalline perovskite film usually exhibit much higher LoD up to hundreds of $\text{nGy}_{\text{air}} \text{ s}^{-1}$, which is similar to the reported LoD value

in our work. Note that our CMOS detector is also fabricated with thick polycrystalline perovskite film rather single crystal perovskite.

Table. List of key parameters of direct conversion X-ray detectors made of metal halide perovskite materials.

Materials	Growth method	Device structure	$\mu\tau$ product (cm ² V ⁻¹)	Sensitivity (μC Gy ⁻¹ cm ⁻²)	Detection limit (nGy s ⁻¹)	Ref
MAPbBr ₃ single crystal	Solvent evaporation/ITC method	Si/PVK/C ₆₀ /BCP/Au	4.0×10 ⁻³	21000	<100	1
MAPbBr _{2.94} Cl _{0.06} single crystal	ITC method	Cr/C ₆₀ /BCP/PVK/Cr	1.8×10 ⁻²	84000	7.6	2
MAPbI ₃ single crystal	Inverse temperature space-confined method	Au/PVK/Au	/	≈710000	1.5	3
GAMAPbI ₃ single crystal	ITC method	Ga/PVK/Au	1.3×10 ⁻²	23000	16.9	4
CsPbI ₃ single crystal	Cooling HI-based precursor solution	Au/PVK/Au	3.63 × 10 ⁻³	2370	219	5
(F-PEA) ₂ PbI ₄ single crystal	Cooling GBL-based precursor solution	Au/PVK/C ₆₀ /BCP/Cr	5.1×10 ⁻⁴	3402	23	6
MAPbBr ₃ single crystal	Antisolvent diffusion	Au/PVK/C ₆₀ /BCP/Ag or Au	1.2×10 ⁻²	80	500	7
Cs ₂ AgBiBr ₆ single crystal	Cooling HBr-based precursor solution	Au/PVK/Au	6.3 × 10 ⁻³	105	59.7	8

Cs ₂ AgBiBr ₆ single crystal	Cooling HBr- based precursor solution	Au/PVK/Au	5.95 × 10 ⁻³	1974	45.7	9
MA ₃ Bi ₂ I ₉ single crystal	Seed-crystal- assisted constant- temperature evaporation	Au/PVK/Au	2.8 × 10 ⁻³	10620	0.62	10
Cs ₂ AgBiBr ₆ Wafer	Hot tableting	Au/PVK/Au	5.51 × 10 ⁻³	250	95.3	11
CsPbBr ₃ film	Hot-press	FTO/PVK/Au	1.32× 10 ⁻²	55684	215	12
CsPbBr ₃ film	Scalable melt processing	Ga/PVK/FTO	/	1450	500	13
MaPbI ₃ film	Solution process	ITO/PVK/Au	6.8 × 10 ⁻⁴	550	67.0	14

1. Wei, W. et al. Monolithic integration of hybrid perovskite single crystals with heterogenous substrate for highly sensitive X-ray imaging. *Nature Photonics* 11, 315-321, doi:10.1038/nphoton.2017.43 (2017).
2. Wei, H. et al. Dopant compensation in alloyed CH₃NH₃PbBr_(3-x)Cl_(x) perovskite single crystals for gamma-ray spectroscopy. *Nature Materials* 16, 826-833, doi:10.1038/nmat4927 (2017).
3. Song, Y. et al. Atomistic Surface Passivation of CH₃NH₃PbI₃ Perovskite Single Crystals for Highly Sensitive Coplanar-Structure X-Ray Detectors. *Research (Wash D C)* 2020, 5958243, doi:10.34133/2020/5958243 (2020).
4. Huang, Y. et al. A-site Cation Engineering for Highly Efficient MAPbI₃ Single-Crystal X-ray Detector. *Angew Chem Int Ed Engl* 58, 17834-17842, doi:10.1002/anie.201911281 (2019).
5. Zhang, B. B. et al. High-Performance X-ray Detection Based on One-Dimensional Inorganic Halide Perovskite CsPbI₃. *Journal of Physical Chemistry Letters* 11, 432-437, doi:10.1021/acs.jpcllett.9b03523 (2020).
6. Li, H. et al. Sensitive and Stable 2D Perovskite Single-Crystal X-ray Detectors Enabled by a Supramolecular Anchor. *Advanced Materials* 32, e2003790, doi:10.1002/adma.202003790 (2020).

7. Wei, H. et al. Sensitive X-ray detectors made of methylammonium lead tribromide perovskite single crystals. *Nature Photonics* 10, 333-339, doi:10.1038/nphoton.2016.41 (2016).
8. Pan, W. et al. Cs₂AgBiBr₆ single-crystal X-ray detectors with a low detection limit. *Nature Photonics* 11, 726-732, doi:10.1038/s41566-017-0012-4 (2017).
9. Yin, L. et al. Controlled Cooling for Synthesis of Cs₂AgBiBr₆ Single Crystals and Its Application for X-Ray Detection. *Advanced Optical Materials* 7, doi:10.1002/adom.201900491 (2019).
10. Zheng, X. et al. Ultrasensitive and stable X-ray detection using zero-dimensional lead-free perovskites. *Journal of Energy Chemistry* 49, 299-306, doi:10.1016/j.jechem.2020.02.049 (2020).
11. Yang, B. et al. Heteroepitaxial passivation of Cs₂AgBiBr₆ wafers with suppressed ionic migration for X-ray imaging. *Nature Communications* 10, 1989, doi:10.1038/s41467-019-09968-3 (2019).
12. Pan, W. et al. Hot-Pressed CsPbBr₃ Quasi-Monocrystalline Film for Sensitive Direct X-ray Detection. *Advanced Materials* 31, 1904405, doi:https://doi.org/10.1002/adma.201904405 (2019).
13. Matt, G. J. et al. Sensitive Direct Converting X-Ray Detectors Utilizing Crystalline CsPbBr₃ Perovskite Films Fabricated via Scalable Melt Processing. *Advanced Materials Interfaces* 7, doi:10.1002/admi.201901575 (2020).
14. Xia, M. et al. Compact and Large-Area Perovskite Films Achieved via Soft-Pressing and Multi-Functional Polymerizable Binder for Flat-Panel X-Ray Imager. *Advanced Functional Materials* 32, 2110729, doi:https://doi.org/10.1002/adfm.202110729 (2022).

Moreover, it is found that the LoD value strongly depends on the electric field intensity, see the following measurements for more details. Specifically, the LoD is 102 nGy_{air} s⁻¹ at 40 V mm⁻¹, 245 nGy_{air} s⁻¹ at 60 V mm⁻¹, and 321 nGy_{air} s⁻¹ at 80 V mm⁻¹, respectively.

The lower limit-of-detection (LoD) response curve of the X-ray detector under various biases: 40V/mm, 60V/mm and 80V/mm.

The above dependency was also reported previously (Advanced Functional Materials, 2022, 2110729) and shows similar trend with our outcome. As reported, the LoD of the detector is 67 nGy_{air} s⁻¹ (SNR = 2.94) at 5 V mm⁻¹, 162 nGy_{air} s⁻¹ (SNR = 2.81) at 10 V mm⁻¹, 317.5 nGy_{air} s⁻¹ (SNR = 5.09) at 50 V mm⁻¹, and 317.5 nGy_{air} s⁻¹ (SNR = 3.16) at 100 V mm⁻¹. Note that the increased bias voltage will increase the device sensitivity but decrease the detection limit value. Please see the revised manuscript on Page 5.

4. In addition, the value of LoD in this work is estimated by the signal-to-noise ratio (Fig. 3e), but the X-ray dose used in the test differs from the estimated value by more than two

orders of magnitude. Please provide more experimental data tested at low X-ray doses as support to increase credibility.

Authors' response: Thank you. We truly appreciate this comment. We measured the detector responses under low X-ray dose of $102 \text{ nGy}_{\text{air}} \text{ s}^{-1}$ at 40 V/mm , see the results below. As seen, the photocurrent was more than three times larger than the noise level, i.e., $\text{SNR}=3.6$, which agrees well with the estimated LoD from the utilized SNR approach.

Moreover, new measurements were performed under lower X-ray dose levels, which are ten times less than used in our previous experiments. The main reason why we did not perform these measurements at even less dose levels is that we want to keep the measurements with higher experimental confidence, i.e., smaller noise fluctuation. By doing so, the predicted LoD values from the linear regression approach would contain least uncertainty. Please see the revised manuscript on Page 5 and Supplementary Figure S17.

The lower limit-of-detection (LoD) response curve of the X-ray detector under various biases: 40V/mm, 60V/mm and 80V/mm.

5. This detector can obtain a fast signal readout speed of 300 fps in this work. The response time of a single-pixel detector should be provided. Moreover, relevant data or reference on the performance of circuits and devices that enable rapid acquisition is also required to be provided.

Authors' response: Thank you. The temporal response of the CsPbBr₃ X-ray detector to square-wave X-ray is measured. As shown below, it has a very quick response with rise-time of 2.5 ms and fall-time of 2.8 ms, which are shorter than the detector readout speed of about 3.3 ms (corresponds to 300 fps).

By the way, the rise-time is defined as the time needed for the photocurrent goes up to 90% of its peak from zero. Similarly, the fall-time is defined as the time needed for the photocurrent goes down to 10% of its peak from the peak. Please see the revised manuscript on Page 6 and Supplementary Figure S25.

Temporal response of the CsPbBr₃ detector to X-ray.

For the used device circuit, the readout speed that is larger than 300 fps has been tested (M.An, C.Chen, C.Gao, M.Han, R.Ji, X.Li, Y.Mei, Q.Sun, X.Sun, K.Wang, L. Xiao, P.Yang, W.Zhou, A low-noise CMOS pixel direct charge sensor, Topmetal-II-, Nucl. Instrum. Methods Phys. Res. Sect.A: Accel. Spectrom. Detect. Assoc. Equip. 810(2016) 144–150, <http://dx.doi.org/10.1016/j.nima.2015.11.153>). Specifically, a 7.8125 MHz clock was supplied to the Scan Module to drive the analog multiplexing. Under such clock frequency, each pixel occupies 128 ns in the analog output of the array, and it takes 0.6636 ms to scan all of the 72×72 pixels in one frame. In other words, each pixel is sampled once every 0.6636 ms, and each sampling lasts 128 ns. Alpha particle induced charge tracks are identified in these time dependent images. A set of images of a single track is shown in Fig.8 of the paper (see below) with a time interval of 3.3 ms. For more details, please take a look at the above paper.

Time slices of a charge track generated by an alpha particle from an ^{241}Am source ionizing ambient air. Time progresses from t_0 to t_5 at equal interval. The time between consecutive images is about 3.3 ms. (Fig. 8, from <http://dx.doi.org/10.1016/j.nima.2015.11.153>)

6. The array size and pixel density are very low (72×72 pixels, $6 \times 6 \text{ mm}^2$). This overshadows advantages of the CMOS array.

Authors' response: Thank you. We agree that the current CMOS array used in this demonstration study is relatively small. However, please be aware that X-ray indirect detectors based on CMOS arrays with areas of approximately $300 \text{ mm} * 300 \text{ mm}$ are already available in the market (Detector Model: AXIOS-3030, Teledyne Dalsa, Canada, <https://www.teledynedalsa.com/en/products/imaging/medical-x-ray-detectors/axios/>). In other words, there have no technical challenges to fabricate large area CMOS for semiconductor FAB to assemble a large area perovskite CMOS detector. Unfortunately, we do not have enough funding support to fabricate that kind of large area CMOS at present. With the purpose of mainly demonstrating the feasibility of assembling direct perovskite detector and achieving advanced X-ray imaging performance, we chose to use a small area CMOS array with affordable development cost. Admittedly, we are very interested in developing large area CMOS array chip and validate its performance in the future.

7. The author mentions that “the stimulated electrons inside the perovskite film were nicely confined within a single pixel by the electric field.”. In my opinion, the carriers generated in adjacent perovskite pixels are more likely to be attracted under larger electric fields (80 V/mm) because perovskite materials have long carrier lifetimes and carrier diffusion lengths.

Authors' response: Thank you. We are sorry for this description. We agree with the reviewer that the carriers generated in adjacent perovskite pixels are more likely to be attracted, especially under high electric fields. To avoid misunderstanding, we have removed this sentence. Please see the revised manuscript on Page 2.

8. *Why was carbon paste used as the electrode material?*

Authors' response: Thank you. According to some previous studies (Matter 2021, 4, 942; Advanced Science 2021, 8, 2102730), the carbon paste is a good candidate material for making the top electrode of X-ray detector. As demonstrated, carbon paste has high stability, low X-ray absorption coefficient and good conductivity. In addition, the carbon electrode can be easily prepared through silk-screen printing, which is consistent with the fabrication procedure of the CsPbBr₃ thick film.

Device configuration of the perovskite X-ray detectors with top carbon electrode.

9. *In Fig. 3b, why does the photocurrent rise first and then fall?*

Authors' response: Thank you. Ideally, the measured photocurrent response of the X-ray detector should in a perfect square shape. However, various material defects such as vacancies, interstitial defects and lattice distortion, and accumulation of ions at the interface may affect the charge generation, transport and extraction. As a result, the ideal square shape would be distorted due to the mixed electronic–ionic nature of perovskites and the presence of some defects, leading to an overshooting peak in the measured photocurrent response curve. In fact, this phenomenon is quite common, and have been observed in many reports. Some of the previous results containing rising spikes are shown below.

(Nature Photonics, 2017, 11, 436-440.)

(Advanced Materials, 2021, 33, 2103078)

(Light: Science & Applications, 2022, 11, 105)

(Nature Photonics, 2022, 16, 575-581.)

Photocurrent responses of perovskite X-ray detector in literature. As seen, they all have rising spikes.

Honestly, we do not have a clear explanation to such phenomenon at present. We noticed that this phenomenon was partially explained in a previous report (Advanced Electronic Materials, 2023, 9, 2300226) for visible light irradiation. Specifically, authors used a three-step carrier transportation model to explain the photocurrent behavior of the perovskites based photodetector. As shown in the following figure, the planar perovskite photodetector is divided into three stacks: surface layer, surface-bulk transition layer, and bulk region. With light excitation, the surface defects were cured by the injected charges within a few milliseconds. Consequently, the surface–bulk transition layer plays an important role in transporting the photocurrent, thus the probability of charge recombination in the surface and surface–bulk transition layers was enhanced. In turn, the photocurrent loss with time (zone B) under green irradiation could be attributed to the combined effect of charge

recombination and ions accumulation due to the increase in Schottky barrier height between metal electrode and perovskite. The ion accumulation depended on the intensity of the electric field and increased with increasing the electric field. In conclusion, the spectral line shape of the photocurrent response is closely related with the irradiance light intensity and applying bias.

(a) Normalized photocurrents of perovskites photodetector. (b) Schematic diagram of the three step carrier transfer model under light emission (Advanced Electronic Materials, 2023, 9, 2300226).

In our opinions, we believe the above validation experiments provide a viable explanation to understand the rising peaks measured on the photocurrent curve. To demonstrate, we further investigated the photocurrent behavior of our CsPbBr₃ X-ray detector at various potential bias and X-ray dose rates. As seen, the overall trends are similar as observed in literature. For instance, the rising of the photocurrent response (normalized by the maximum) increases as the applied bias increases. Moreover, the rising of the photocurrent response (normalized by the maximum) increases as the X-ray exposure dose rate increases.

Photocurrent responses of X-ray detector at different bias voltages and dose rates.

10. In Fig. 3e, there is a typo, “Signal to niose ratio”.

Authors’ response: Thank you. We are sorry for this typo, it should be “Signal to noise ratio”. Please see the revised Fig. 3e.

Reviewer #2 (Remarks to the Author):

The authors present an interesting X-ray detector, combining the all inorganic CsPbBr₃ perovskite and complementary metal-oxide-semiconductor (CMOS) pixels. Although the device does not exhibit record sensitivities or mobility lifetime products, they are among the best in perovskite-based devices. Moreover, they show exceptional 2D and 3D imaging utilizing the CMOS array, lowering the needed dose-rate and increasing spatial resolution, all at a fast readout speed. They definitely emphasize the potential of perovskite when combined with CMOS for medical applications. The study is well designed and presented. Moreover, the manuscript is nicely written and reads well. I have a few questions, that if answered, I would recommend publication in Nature Communications.

Authors' response: Thank you.

1) When discussing the screen-printing of CsPbBr₃, could you elaborate a bit more on the intermediate phase, which is common in perovskite in similar printing methods. (<https://doi.org/10.1021/acsnano.0c07993>)

Authors' response: Thank you. We completely agree with the reviewer that the intermediate phases from CsPbBr₃ paste to CsPbBr₃ thick film is important. To address this comment, detailed experiments and results are presented. In particular, the XRD patterns of PbBr₂-DMSO and PbBr₂-DMF are investigated. Compared with original PbBr₂ powder, as seen, occupation of PbBr₂ and DMSO or DMF varies significantly (ACS Nano, 2021, 15, 4077-4084). The CsPbBr₃ paste prepared with such binary solvents at a volume ratio of 7:3 (V_{DMSO}:V_{DMF}) presents the main intermediate phase of Cs(DMSO)PbBr₃. Under solvent evaporation, the nucleation and crystal growth of CsPbBr₃ leads to an increase of the diffraction signals of CsPbBr₃ at stage 1. Meanwhile, the diffraction signals corresponds to the intermediate phase of Cs(DMSO/DMF)PbBr₃ shows up. With further annealing, the main intermediate phase changes to Cs(DMSO/DMF)PbBr₃ at stage 2, suggesting that the thermal stability of Cs(DMSO/DMF)PbBr₃ is much better than that of Cs(DMSO)PbBr₃. Eventually, the Cs(DMSO/DMF)PbBr₃ decomposes and CsPbBr₃ thick film forms at stage 3 with continuous annealing.

In addition, thanks for bring the important work into our attention, and we have added it in the reference in the revised manuscript. Please see the revised manuscript on Page 4 and Supplementary Fig. S11-S13.

XRD patterns of PbBr_2 powder and its adduct with DMSO or DMF, and the crystallization of CsPbBr_3 thick film from CsPbBr_3 paste.

2) Indicate the light source used in your mobility lifetime calculations.

Authors' response: Thank you. For the mobility lifetime product calculation, the photoconductivity of the CsPbBr_3 thick film was measured under illumination of a 460 nm LED blue light source. The blue light was absorbed by the surface of the CsPbBr_3 thick film, and the excited charges were collected by electrodes at bias across the entire thick perovskite film. Please see the revised manuscript on Page 4.

3) You mention the maximum dark to photocurrent ratio is 2000, however this is at a position of a dip in the dark current. Why is the dip? Is it reproducible? An average ratio would be better to use. (Fig. 3a)

Authors' response: Thank you. In this study, a SnO_2 electron transport layer (ETL), which is able to partially prevent the charges injection from electrode to perovskite, was inserted between the ITO and the CsPbBr_3 film to accelerate the extraction of photo-generated electrons and inhibit charge injection. As a result, the rectifying behavior of the Schottky contact between perovskites and electrodes would vary as the added bias voltage changes. For example, the measured J-V curve shows a rectifying behavior at low applying bias, and the rectifying behavior vanishes as the bias voltage increases. We guess such behavior change results a small dip (discontinuity) on the dark current curve.

In order to investigate the repeatability of this phenomenon, the dark current performance of ten X-ray detectors was tested, and results are shown below. As seen, the measured dark current curves of all the ten samples exhibit the dip behaviors.

Device configuration of the CsPbBr_3 X-ray detector.

We agree with the reviewer that the averaged photo current to dark current ratio would be a better choice to statistically represent the photo response of the perovskite material. On average, the photo current to dark current ratio is 219.86, with a maximum ratio up to 1989.32 at the electric field of 24 V/mm . Please see the revised manuscript on Page 5.

(Left) The ratio of photo current to dark current with respect to the electric field; (Right) the histogram distribution of the ratio of photo current to dark current.

4) You varied your dose-rates from higher to lower, did you do any reproducibility checks and have you started with lower to higher dose rates. Can you explain the initial drop in the photocurrent, which eventually starts to rise as seen in Figure 3f. What can be done to keep this current stable for application, even in the cycling test the baseline is moving? I believe this should be discussed a bit more in detail.

Authors' response: Thank you. The photocurrent responses were measured separately under two opposite conditions: one goes from lower X-ray dose rate up to higher X-ray dose rate, and the other goes from higher X-ray dose rate down to lower X-ray dose rate. Results are presented below. As seen, the fabricated CsPbBr₃ detector can work well under both conditions.

The X-ray detector photocurrent response measured (left) from lower to higher X-ray dose rates, and (right) from higher to lower X-ray dose rates.

We guess the initial drop of photocurrent in Fig. 3f may be caused by the ionic nature of perovskite material. Unfortunately, we are sorry that we do not have a clear explanation to it now. Under X-ray irradiation, we guess the surface defects were cured by the injected charges, leading to an initial overshoot of the photocurrent. After that, the photocurrent gradually decreases and becomes saturated. During a long-term working period, the photocurrent and dark current baseline of the detector increase gradually. This may be caused by the field-driven ion-migration induced polarization of the perovskite thick film (Haruyama, J., Sodeyama, K., Han, L. & Tateyama, Y. First-Principles Study of Ion Diffusion in Perovskite Solar Cell Sensitizers. *Journal of the American Chemical Society* 137, 10048-10051). To improve, several approaches can be utilized to minimize the variations of photocurrent in the future. For example, optimizing the interface material between the perovskite film and the electrode, optimizing the added external voltage bias, and optimizing the perovskite material with least ionic nature. For example, the low dimension perovskites with suppressed ion-migration effect and high resistivity are demonstrated to be beneficial in stabilizing the output current (Zhuang, R. et al. Highly sensitive X-ray detector made of layered perovskite-like $(\text{NH}_4)_3\text{Bi}_2\text{I}_9$ single crystal with anisotropic response. *Nature Photonics* 13, 602-608). In addition, surface passivation is also a promising strategy to heal the surface defect of perovskite thick film and suppress

the ion-migration (Song, Y. et al. Elimination of Interfacial-Electrochemical-Reaction-Induced Polarization in Perovskite Single Crystals for Ultrasensitive and Stable X-Ray Detector Arrays. 33, 2103078). We believe these optimizations could help to dramatically enhance the detection performance. Please see the revised manuscript on Page 5.

5) Did you characterize your sample after these long-term tests (cycling or under constant operation/irradiation), any changes or degradation?

Authors' response: Thank you. We agree with the reviewer that device stability is critical for continuous radiation exposures in real applications. To test, the reliability of the CsPbBr₃ thick film X-ray detectors under constant irradiation with electric field of 10 V mm⁻¹ and dose rate of 4886 μGy s⁻¹ are tested for 60000 seconds. The accumulated radiation dose to this CsPbBr₃ detector exceeds 290 Gy_{air}.

As shown in the plot, overall, the photocurrent slowly increases after a very long testing period. However, such increase over a short time period, e.g., several seconds or several tens of seconds, is essentially negligible. As a consequence, such smooth and slight variations over a short time period would not bring too much troubles for real medical imaging applications. This is because most of the medical imaging tasks merely last for a very short time, for instance, several seconds (or one minute at most). For most of the times, the tube output also needs time to stabilize its output. Usually, such tube output gradually increases at the very beginning several seconds. To deal with such intensity variations, technically, proper corrections/calibrations are employed in practice to make the measured signals become uniform. To this end, we think the slight variations over a short time period of the detector would bring too much trouble.

To improve, several approaches can be utilized to minimize the variations of photocurrent in the future. For example, optimizing the interface material between the perovskite film and the electrode, optimizing the added external voltage bias, and optimizing the perovskite material with least ionic nature. We believe these optimizations could help to dramatically enhance the detection performance.

Long-time responses of photocurrent of the CsPbBr₃ detector under constant irradiation.

After the long-term testing (≥ 60000 s), the carbon electrode on the detector was removed to test the irradiation stability of the CsPbBr₃ thick film. Compared with the original CsPbBr₃ thick film (before irradiation), as seen, the long-term tested CsPbBr₃ thick film exhibits comparable XRD pattern, indicating the crystal structure of the CsPbBr₃ perovskites is fairly stable under constant X-ray irradiation. Moreover, the CsPbBr₃ perovskite remains uniform and emit strong green luminescence after long-term irradiation. Please see the revised manuscript on Page 5 and Supplementary Fig. S19.

XRD and PL patterns of the original and long-term irradiated CsPbBr₃ film.

Reviewer #3 (Remarks to the Author):

This manuscript is written about X-ray detectors with CMOS array and has been systematically studied. Therefore, the reviewer believes that the manuscript is suitable for publication in Nature Com., but he believes that further consideration is necessary for the following points.

Authors' response: Thank you.

1. In the background, the authors state that "the pixel size of CMOS arrays can be easily made less than 5 μm ". However, the pixel size of the actually utilized CMOS array is about 80 μm . At the moment, this seems to be the limit when considering the crystal size, etc., but if there is any other reason for using this size CMOS array, please mention it.

Authors' response: Thank you. We agree that the perovskite crystal size is one of the key factors that may impact the selection of the pixel size of CMOS arrays. The dominant reason why we chose 80 μm pixel sized CMOS array is because most of the current commercial CMOS detectors used for medical X-ray and CT imaging applications have pixel sizes range from 50 μm to 150 μm . In fact, such a pixel dimension also well balances the total imaging area (detector size) and the signal read out speed of an X-ray CMOS detector. Since our current research interests are focused on validating the medical X-ray imaging performance of perovskite based direct X-ray detector, therefore, the pixel size of the CMOS array was designed to have a moderate dimension, i.e., 80 μm .

For optical and biomedical optical imaging applications, essentially, the pixel size of the CMOS sensor array can be made less than 5 μm . Under this conditions, usually, the size of the entire sensor becomes quite small. For example, a 2048*2048 dimensioned CMOS array with 5 μm element size correspond to an area of 10.24 mm * 10.24 mm. Apparently, this is not suitable for most of the medical X-ray and CT imaging applications.

We have added this reason into the revised manuscript, please see it on Page 2.

2. The advantage of CMOS arrays, suppression of leakage dark current, is also mentioned in the background. This is interesting for readers. Please describe about the dark current results for the CMOS array (with special electrical circuits) and the TFT array.

Authors' response: Thank you. A Charge Sensitive Amplifier (CSA) with a suppressing dark current circuit used in the CMOS array is shown below. The dark current from the detector is absorbed by the transistor M1. The current generated by M1 is automatically adjusted according to the feedback circuit which is composed of a resistor, a capacitor and an open-loop amplifier to match the dark current of the detector. Hence, the baseline of the CSA_OUT node can be obtained with the same level of the CSA_VREF node. As a consequence, the output dynamic range of the CSA is not affected by the dark current.

A Charge Sensitive Amplifier (CSA) with a suppressing dark current circuit used in the CMOS array.

By referring to literature (Sheth, Niraj Milan, Ali Uneri, Patrick A. Helm, Wojciech Zbijewski, and Jeffrey H. Siewerdsen. "Technical assessment of 2D and 3D imaging performance of an IGZO-based flat-panel X-ray detector." Medical physics 49, no. 5 (2022): 3053-3066), the dark current response of CMOS and TFT is 200 and 2000, respectively. Compared to CMOS, apparently, TFT has higher dark current (circuit readout

noise) and thus is not suitable for ultra-low dose X-ray imaging. On the contrary, CMOS is more appropriate for ultra-low dose X-ray imaging due to its low dark current response. Please see the revised manuscript on Page 2.

3. (i) Regarding the J-V curve in Fig.3.a, (i) Dark current is usually determined by the resistivity of the film and the contact with the electrode. Light current is determined by detection sensitivity and quantum efficiency in addition to dark current. So each should have a different electric field dependence, but why do they look nearly translated?

Authors' response: Thank you. Visually, the measured J-V curves of the light current and the dark current are similar. To compare, the J-V curve of the X-ray detector is normalized with respect to its own maximum. Clearly, the responses of the light current and the dark current are very different.

Current density-voltage (J-V) curve of the CsPbBr₃ detector.

Moreover, the ratio of photo current to dark current with respect to the electric field is also plotted. Since the ratio plot contains clear valleys and peaks with various values, therefore, the responses of the light current and the dark current are different.

Current density-voltage (J-V) curve of the CsPbBr₃ detector.

(ii). Related to the previous question, why is the electric field not monotonically decreasing over the -190--170 V/mm range? The reviewer is concerned that the electric field may not simply represent the electric field in the detection layer. (For example, the electric field is applied to the electrode or the interface between the electrode and the detection layer more than expected.

Authors' response: Thank you. We appreciate this comment. We think this is related to the ionic nature of the perovskite film. Usually, the ionic perovskites suffer from the ion-migration effect under large electric field. In particular, the solution-processed perovskite thick film inevitably contains pores and pinholes due to the evaporation of the solvent, and the hot-pressed procedure is able to greatly improve the perovskite film density but can't completely remove the pinholes. Therefore, the pinholes in perovskite film lead to more defects and ion migration path way, causing serious recombination current loss. Moreover, the perovskite film may be polarized and decomposed under high voltage, which inhibits charge transport and thus lead to lower current density.

In one previous report (Advanced Functional Materials 2022, 32, 2110729), the solution processed MAPbI₃ perovskite film also exhibits similar abnormal trend as obtained in our experiments, see the following plot. To mitigate, the authors added trimethylolpropane triacrylate (TMTA) as the binder to work with MAPbI₃ and passivate the grain boundaries in perovskites thick film. As a result, the optimized MAPbI₃/TMTA X-ray detector exhibits

much lower dark current. Most importantly, the abnormal behavior/trend of the current density disappears. Similarly, the measured abnormal behavior/trend of the current density can be mitigated if special additive material is used to modify CsPbBr₃ and passivate the defects inside the CsPbBr₃ thick film.

(Advanced Functional Materials, 2022, 32, 2110729)

The measured dark current density of the MAPbI₃ and MAPbI₃/TMTA thick film based X-ray detector published in literature.

4. Please add measurement conditions (tube voltage, tube current, dose rate) about Fig 3.d.

Authors' response: Thank you. We are sorry for missing such important information. In the revised manuscript, parameters of the X-ray (Rh) source such as the tube voltage (50 kV), tube current (20 uA) and dose rate (160 $\mu\text{Gy}_{\text{air}} \text{s}^{-1}$) are added in Figure 3d.

Plots of sensitivity and SNR of the X-ray detector.

5. For Fig3.e, it is unacceptable to determine the detection limit with this extrapolated line. If this film can detect 1/1000 of the intensity of X-rays that had actually been detected, show that X-rays of that dose can be detected by adjusting the dose rate with an attenuation plate or the like.

Authors' response: Thank you. We truly appreciate this suggestion, and we agree with the reviewer that the previous measurements and calculations (linear fittings) might have potential drawbacks. To mitigate, additional beam filtration was added to lower the dose rate by 10-100 times. This time, the measured LoD value of CsPbBr₃ detector was found to be 102 nGy_{air} s⁻¹ at 40 V mm⁻¹, 245 nGy_{air} s⁻¹ at 60 V mm⁻¹, and 321 nGy_{air} s⁻¹ at 80 V mm⁻¹, respectively.

Particularly, the photocurrent responses at dose rate of 102 nGy_{air} s⁻¹ was measured. Results show that the photocurrent was more than three times higher than the noise level, corresponding to SNR=3.6. To this end, we think the estimation of LoD using the current extrapolation approach is viable. Moreover, since the estimation accuracy of LoD may be impacted by the measured data points, therefore, we think it is necessary to keep the data points having high enough SNRs. Namely, measuring the data points under high dose rates. Please see the revised manuscript on Page 5 and Supplementary Figs. S17.

The lower limit-of-detection (LoD) response curve of the X-ray detector at various bias.

6. As for the experiment on Fig. 4, the reviewer thinks the results are excellent. However, if there is a reason why the result shown is only at 80V/mm, please describe. For example, a measurement of 20 or 40 V/mm, which has a higher SNR, should give a higher-resolution image.

Authors' response: Thank you. This is a great question. During our experiments, the reason why we set the added electric field at 80V/mm rather than other lower values is because we feel like the images obtained at 80V/mm presents the highest quality. The results obtained at different external voltages are shown in Fig. S23, see the results below.

CMOS detector responses at varied bias voltages.

As seen, the image resolution, the signal intensity and the image uniformity all get enhanced as the added bias increases. Roughly, the generated images at 25 V (corresponds to electric field of about 67V/mm) and 30 V (corresponds to electric field of about 80V/mm) look very similar. Based on these experimental results, we set the bias voltage at 30 V during our tests.

We want to point that lowering the electric field might bring degradations to image spatial resolution, especially when the electric field gets too small. We think this is due to the less efficient confinement of electrons inside the perovskite material. For X-ray imaging, we have to balance a lot of parameters at the same time, and cannot merely rely on one single parameter such as SNR or sensitivity. As mentioned by the reviewer, a measurement of 20 or 40 V/mm provides a higher SNR, however, the sensitivity of the perovskite material gets too low to generate sufficient number of electrons. The increase of SNR is due to the reduced dark current. For our device used in this study, we believe the choice of 80/V well

balances the material sensitivity, dark current, and the image quality (we believe this is the most important factor we have to consider).

7. Please briefly explain the necessity of SnO₂ for the methods (device fabrication). The authors can also reference some literature.

Authors' response: Thank you. In our X-ray detector, the SnO₂ film is used as an electron transport layer (ETL) for electron transportation and collection. In addition, the compact SnO₂ film can also separate the CsPbBr₃ film from the bottom metal electrode, which is good for enhancing the device stability.

To fabricate the SnO₂ film, specifically, the colloidal SnO₂ aqueous dispersion (15 wt%) was diluted using deionized water to 5 wt%, and the solution was stirred at room temperature for 2 hours. After that, the obtained SnO₂ solution was spin-coated on ITO/CMOS substrate at 4000 rpm for 30 s, and the sample was annealed at 150 °C for 60 minutes to remove the residual solvent and surfactant achieving complete SnO₂ film.

The reference (Liu, C. et al. Hydrothermally Treated SnO₂ as the Electron Transport Layer in High-Efficiency Flexible Perovskite Solar Cells with a Certificated Efficiency of 17.3%. *Advanced Functional Materials* 29, 1807604, (2019)) has added into the revised manuscript. Please see the revised manuscript on Page 8 (with reference 34).

REVIEWERS' COMMENTS

Reviewer #1 (Remarks to the Author):

The authors have carefully considered the comments and questions from the reviewer. Most of the unclear statements or conclusions in the first version of the manuscript have been clarified by additional figures and discussions. I am satisfied with most of the authors' answers.

However, I have another concern:

Q1: In the latest modified version, the data in Figure 3e seems to be lost. In addition, in the previous version, the minimum detection limit value was 346 nGyair s⁻¹. But under the same condition, the value was changed to 321 nGyair s⁻¹ in the revised manuscript. Please check and explain the reason.

Reviewer #2 (Remarks to the Author):

I would like to thank the authors for clearly answering all my questions and implementing the necessary changes. I can now recommend your article for publication in Nature Communication.

Reviewer #3 (Remarks to the Author):

The reviewers believe that the manuscript has been appropriately revised and is worthy of publication in Nature Communications.

Please make the following minor corrections.

In Fig. 3(e), it seems that data points (plots) are missing. Please add them.

The reviewer recommends that the data points obtained from Fig. S17(c) (Dose rate = 102 and SNR = 3.6) is included in Fig. S17(a).

Title: Dynamic X-ray imaging with screen-printed perovskite CMOS array

Paper ID: NCOMMS-23-31690B

We would like to sincerely thank the reviewers and editors for their time and efforts in reviewing our manuscript for one more time. All comments and suggestions have been addressed carefully. Please find our point-by-point responses below.

Note: the reviewers' comments are in blue italic text, and our point-by-point responses are in plain black text.

Reviewer #1 (Remarks to the Author):

The authors have carefully considered the comments and questions from the reviewer. Most of the unclear statements or conclusions in the first version of the manuscript have been clarified by additional figures and discussions. I am satisfied with most of the authors' answers. However, I have another concern:

Q1: In the latest modified version, the data in Figure 3e seems to be lost. In addition, in the previous version, the minimum detection limit value was $346 \text{ nGy}_{\text{air}} \text{ s}^{-1}$. But under the same condition, the value was changed to $321 \text{ nGy}_{\text{air}} \text{ s}^{-1}$ in the revised manuscript. Please check and explain the reason.

Authors' response: Thank you. We are very sorry that the data points were missed when converting the word file into the pdf file due to unknown reasons. We re-converted the file, and it happened again. To eliminate this mistake, we updated the subplot of Fig. 3e, and this time the data points will not be lost, please see the revised pdf file.

In the revised manuscript, essentially, measurements were made under different low X-ray dose levels with a newly prepared perovskite X-ray detector. As a result, the minimum detection limit value ($321 \text{ nGy}_{\text{air}} \text{s}^{-1}$) was slightly different from the previously estimated minimum detection limit value of $346 \text{ nGy}_{\text{air}} \text{s}^{-1}$. We believe the new measurements and estimations are more accurate.

Reviewer #2 (Remarks to the Author):

I would like to thank the authors for clearly answering all my questions and implementing the necessary changes. I can now recommend your article for publication in Nature Communication.

Authors' response: Thank you for your patient review and comments.

Reviewer #3 (Remarks to the Author):

The reviewers believe that the manuscript has been appropriately revised and is worthy of publication in *Nature Communications*. Please make the following minor corrections.

1. In Fig. 3(e), it seems that data points (plots) are missing. Please add them.

Authors' response: Thank you. We are very sorry that the data points were missed when converting the word file into the pdf file due to unknown reasons. We re-converted the file, and it happened again. To eliminate this mistake, we updated the subplot of Fig. 3e, and this time the data points will not be lost, please see the revised pdf file.

In the revised manuscript, essentially, measurements were made under different low X-ray dose levels with a newly prepared perovskite X-ray detector. As a result, the minimum detection limit value ($321 \text{ nGy}_{\text{air}} \text{ s}^{-1}$) was slightly different from the previously estimated minimum detection limit value of $346 \text{ nGy}_{\text{air}} \text{ s}^{-1}$. We believe the new measurements and estimations are more accurate.

2. The reviewer recommends that the data points obtained from Fig. S17(c) (Dose rate = 102 and SNR = 3.6) is included in Fig. S17(a).

Authors' response: Thank you. According to the reviewer's suggestion, we have added the average value of the measured data points in Supplementary Fig. 17c into the

Supplementary Fig. 17a. Please see the red dot in the revised Supplementary Fig. 17a. Clearly, results demonstrate that the linear estimation of LoD is good enough to use.